# NTK-DFL: Enhancing Decentralized Federated Learning in Heterogeneous Settings via Neural Tangent Kernel

## Abstract

Decentralized federated learning (DFL) is a collaborative machine learning framework for training a model across participants without a central server or raw data exchange. DFL faces challenges due to statistical heterogeneity, as participants often possess different data distributions reflecting local environments and user behaviors. Recent work has shown that the neural tangent kernel (NTK) approach, when applied to federated learning in a centralized framework, can lead to improved performance. The NTK-based update mechanism is more expressive than typical gradient descent methods, enabling more efficient convergence and better handling of data heterogeneity. We propose an approach leveraging the NTK to train client models in the decentralized setting, while introducing a synergy between NTK-based evolution and model averaging. This synergy exploits inter-model variance and improves both accuracy and convergence in heterogeneous settings. Our model averaging technique significantly enhances performance, boosting accuracy by at least 10% compared to the mean local model accuracy. Empirical results demonstrate that our approach consistently achieves higher accuracy than baselines in highly heterogeneous settings, where other approaches often underperform. Additionally, it reaches target performance in 4.6 times fewer communication rounds. We validate our approach across multiple datasets, network topologies, and heterogeneity settings to ensure robustness and generalizability.

## 1 Introduction

Federated learning (FL) is a machine learning paradigm in which multiple clients train a global model without the explicit communication of training data. In most FL scenarios, clients communicate with a central server that performs model aggregation. In the popular federated averaging (FedAvg) algorithm (McMahan et al., 2017), clients perform multiple rounds of stochastic gradient descent (SGD) on their own local data, then send this new weight vector to a central server for aggregation. As FL gains popularity in both theoretical studies and real-world applications, numerous improvements have been made to address challenges, including communication efficiency, heterogeneous data distributions, and security concerns (Sattler et al., 2020; Li et al., 2020; Zhu et al., 2019). To handle the performance degradation caused by data heterogeneity, many works have proposed mitigation for FedAvg (Karimireddy et al., 2020; Li et al., 2020). Notably, some researchers have introduced the neural tangent kernel (NTK), replacing the commonly-used SGD in order to improve the model convergence (Yu et al., 2022; Yue et al., 2022).

Despite these advancements, the centralized nature of traditional FL schemes introduces the possibility for client data leakage, computational bottlenecks at the server, and high communication bandwidth demand (Kairouz et al., 2021). Decentralized federated learning (DFL) has been proposed as a solution to these issues (Martínez Beltrán et al., 2023). In DFL, clients may communicate with each other along an undirected

graph, where each node represents a client and each edge represents a communication channel between clients. While DFL addresses some of the issues inherent to centralized FL, both frameworks grapple with the challenge of statistical heterogeneity across clients. Although mixing data on a central server could readily resolve this issue, privacy concerns and the burden of extensive communication make FL and DFL approaches necessary to address this challenge. This paper focuses on the following research question: How can we design a DFL approach that effectively addresses statistical heterogeneity?

We propose a DFL method that exploits the NTK to evolve weights. We denote this paradigm NTK-DFL. Our approach combines the advantages of NTK-based optimization with the decentralized structure of DFL. The NTK-DFL weight evolution scheme makes use of the communication of client Jacobians, allowing for more expressive updates than traditional weight vector transmissions and improving performance under heterogeneity. Complementing this NTK-based evolution, we utilize a model averaging step that exploits inter-model variance among clients, creating a global model with much better generalization than any local model. We demonstrate that NTK-DFL maintains high performance even under aggressive compression measures. Through reconstruction attack studies, we also analyze how this compression affects data privacy. The contributions of this paper are threefold.

1. The proposed NTK-DFL method achieves convergence with 4.6 times fewer communication rounds than existing approaches in heterogeneous settings. To the best of our knowledge, this is the first work leveraging NTK-based weight evolution for decentralized federated training.

2. The effective synergy between NTK-based evolution and DFL demonstrates superior resilience to data heterogeneity with model averaging.

3. The NTK-DFL aggregated model achieves at least 10% higher accuracy than the average accuracy of individual client models. This aggregated model exhibits robust performance across various network topologies, datasets, data distributions, and compression measures.

## 2 RELATED WORK

**Federated Learning (FL)**   FL was introduced by McMahan et al. (2017) as a machine learning approach that enables training a model on distributed datasets without sharing raw data. It attempts to address key issues such as data privacy, training on decentralized data, and data compliance for more heavily regulated data (e.g., medical imaging) (Zhang et al., 2021). Despite its advantages, the centralized topology of FL introduces several challenges. These include potential privacy risks at the central server, scalability issues due to computational bottlenecks, and high communication overhead from frequent model updates between clients and the server (Mothukuri et al., 2021).

**Decentralized Federated Learning (DFL)**   DFL aims to eliminate the need for a central server by connecting clients in a fully decentralized topology. Sun et al. (2023) adapted the FedAvg approach of multiple local SGD iterations to the decentralized setting, leveraging momentum to improve model convergence and weight quantization to reduce total communication cost. Our NTK-DFL method may be viewed as building on this foundation, using the neural tangent kernel for more effective weight updates. Dai et al. (2022) proposed a method of DFL where each client possesses their own sparse mask personalized to their specific data distribution. Shi et al. (2023) employed the sharpness-aware minimization optimizer to reduce the inconsistency of local models, whereas we tackle this issue through per-round averaging and final model aggregation. DFL approaches can aim to train one global model, such as the case of many hospitals training a model for tumor classification with local, confidential images (Shiri et al., 2022). They may also aim to train a personalized model for each client in order to perform better on the local data distribution. For example, different groups of mobile phone users may use different words or emojis and would benefit from a personalized model (Tan et al., 2023). Our method focuses on training a high-performing global model

that generalizes well across all clients, offering improved convergence and resilience to data heterogeneity compared to existing DFL approaches.

**Neural Tangent Kernel (NTK)**   NTK has primarily been used for the analysis of neural networks (Golikov et al., 2022), though it has recently seen use in the training of neural networks for FL (Yue et al., 2022). Introduced by Jacot et al. (2018), it shows that the evolution of an infinitely wide neural network converges to a kernelized model. This approach has enabled the analytical study of models that are well approximated by this infinite width limit (Liu et al., 2020). The NTK has also been extended to other model types, such as the recurrent neural network (Alemohammad et al., 2021) and convolutional neural network (Arora et al., 2019). We instead use the linearized model of the NTK approximation as a tool for weight evolution. Some studies have explored the integration of NTKs with FL. For instance, Huang et al. (2021) applied the NTK analysis framework to study the convergence properties of FedAvg, while Yu et al. (2022) extended NTK applications beyond theoretical analysis by training a convex neural network. Moreover, Yue et al. (2022) replaced traditional SGD-based optimization with NTK-based evolution in a federated setting, where clients transmit Jacobian matrices to a central server that performs weight updates using the NTK.

# 3   PROPOSED METHOD: NTK-BASED DECENTRALIZED FEDERATED LEARNING

## 3.1   PROBLEM STATEMENT

We begin with a brief overview of centralized FL. The goal of centralized FL is to train a global model $\boldsymbol{w}$ across $M$ clients with their private, local data $\mathcal{D}_i = \{(\boldsymbol{x}_{i,j}, \boldsymbol{y}_{i,j})\}_{j=1}^{N_i}$, where $N_i$ is the number of training examples of the $i$th client. FL algorithms aim to numerically solve the sample-wise optimization problem of $\min_{\boldsymbol{w}} F(\boldsymbol{w})$, where $F(\boldsymbol{w}) = \frac{1}{M} \sum_{i=1}^{M} N_i F_i(\boldsymbol{w})$ and $F_i(\boldsymbol{w}) = \frac{1}{N_i} \sum_{j=1}^{N_i} \ell(\boldsymbol{w}, \boldsymbol{x}_j, \boldsymbol{y}_j)$, where $M$ denotes the number of clients. In the decentralized setting, an omnipresent global weight $\boldsymbol{w}$ is not available to clients in each communication round. Rather, each client possesses their own model $\boldsymbol{w}_i$ that is trained in the update process. Following related DFL work (Shi et al., 2023; Sun et al., 2023), we seek a global model $\boldsymbol{w}$ that benefits from the heterogeneous data stored locally across clients and generalizes better than any individual client model. A global or aggregated model may take the form $\boldsymbol{w} = \frac{1}{N} \sum_{i=1}^{M} N_i \boldsymbol{w}_i$, where $N = \sum_{i=1}^{M} N_i$.

**Notation**   Formally, we have a set of clients $\mathcal{C} = \{1, \ldots, i, \ldots, M\}$. Each client is initialized with its weight $\boldsymbol{w}_i^{(0)} \in \mathbb{R}^d$, where $d$ is the size of the parameter vector and the superscript in $\boldsymbol{w}_i^{(0)}$ denotes the initial communication round. Model training is done in a series of communication rounds denoted $k \in \{1, 2, ..., K\}$. Let the graph at round $k$ be $\mathcal{G}^{(k)} = (\mathcal{C}, E^{(k)})$, where $E^{(k)}$ is the set of edges representing connections between clients. Furthermore, the neighborhood of client $i$ at round $k$ is denoted $\mathcal{N}_i^{(k)} = \{j \mid (i, j) \in E^{(k)}\}$. This graph is specified before each communication round and can take an arbitrary form.

## 3.2   COMMUNICATION PROTOCOL

In the following sections, we present the NTK-DFL paradigm (Figure 1). We describe the key components of the algorithm, including the communication protocol and weight evolution process.

**Per-round Parameter Averaging**   At the beginning of each communication round $k$, each client $i$ both sends and receives weights. Every client sends their model $\boldsymbol{w}_i^{(k)}$ to all neighbors $j \in \mathcal{N}_i^{(k)}$. Simultaneously, $i$ receives the weight vectors $\boldsymbol{w}_j^{(k)}$ from all neighbors $j \in \mathcal{N}_i^{(k)}$. Each client then aggregates its own weights along with the neighboring weights to form a new weight as follows:

$$\bar{\boldsymbol{w}}_i^{(k)} = \frac{1}{N_i + \sum_{j \in \mathcal{N}_i^{(k)}} N_j} \Big( N_i \boldsymbol{w}_i^{(k)} + \sum_{j \in \mathcal{N}_i^{(k)}} N_j \boldsymbol{w}_j^{(k)} \Big). \tag{1}$$

The client must then send this aggregated weight $\bar{\boldsymbol{w}}_i^{(k)}$ back to all neighbors $j \in \mathcal{N}_i^{(k)}$. This step enables each client to construct a local NTK, comprised of inner products of Jacobians from both neighboring clients and their own Jacobians. See Algorithm 2 in Appendix A for a detailed description of this process.

**Local Jacobian Computation** At this point, each client possesses its own aggregated weight $\bar{\boldsymbol{w}}_i^{(k)}$ as well as an aggregated weight $\bar{\boldsymbol{w}}_j^{(k)}$ for each of their neighbors $j \in \mathcal{N}_i^{(k)}$. The clients use these weights $\bar{\boldsymbol{w}}_j^{(k)}$ and local data $\mathbf{X}_i$ to compute the Jacobian of $f(\mathbf{X}_i; \bar{\boldsymbol{w}}_j^{(k)})$ with respect to the neighboring model parameters $\bar{\boldsymbol{w}}_j^{(k)}$ and their local data $\mathbf{X}_i$. We can denote this neighbor-specific Jacobian as

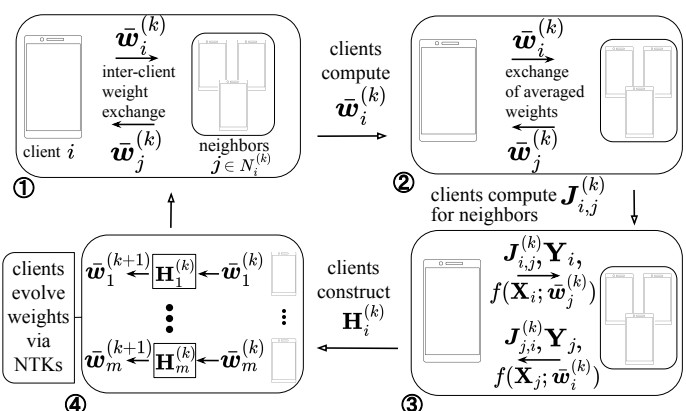

$$\boldsymbol{J}_{i,j}^{(k)} \equiv [\nabla_{\boldsymbol{w}} \boldsymbol{f}(\mathbf{X}_i; \bar{\boldsymbol{w}}_j^{(k)})]^\top. \quad (2)$$

For a given client, the gradient is taken with respect to its neighbor's aggregated weight $\bar{\boldsymbol{w}}_j^{(k)}$, but the function is evaluated on the client's local data $\mathbf{X}_i$. Each client sends every neighbor their respective Jacobian tensor $\boldsymbol{J}_{i,j}^{(k)}$, true label $\mathbf{Y}_i$,

Figure 1: NTK-DFL process: ① Clients exchange weights, ② Average weights with neighbors, ③ Compute and exchange Jacobians, labels, and function evaluations, ④ Construct local NTK and evolve weights [Eq. (5)]. This decentralized approach enables direct client collaboration and NTK-driven model evolution without a central server.

and function evaluation $\boldsymbol{f}(\mathbf{X}_i; \bar{\boldsymbol{w}}_j^{(k)})$. Note the order of the indices in the Jacobian: the client sends $\boldsymbol{J}_{i,j}^{(k)}$, an evaluation on the client's data and the neighbor's weights. In contrast, the client receives $\boldsymbol{J}_{j,i}^{(k)}$ from each of its neighbors, an evaluation on the neighbor's data and the client's weights. Algorithm 3 in Appendix A describes this process.

## 3.3 WEIGHT EVOLUTION

After all inter-client communication is completed, the clients begin the weight evolution phase of the round (see Algorithm 4 in Appendix A). Here, all clients act in parallel as computational nodes. Each client possesses their own Jacobian tensor $\boldsymbol{J}_{i,i}^{(k)}$ as well as their neighboring Jacobian tensors $\boldsymbol{J}_{j,i}^{(k)}$ for each $j \in \mathcal{N}_i^{(k)}$.

We denote the tensor of all Jacobian matrices possessed by a client at round $k$ as $\boldsymbol{\mathcal{J}}_i^{(k)}$, which is composed of matrices from the set $\{\boldsymbol{J}_{i,i}^{(k)}\} \cup \{\boldsymbol{J}_{j,i}^{(k)} \mid j \in \mathcal{N}_i^{(k)}\}$ stacked along a third dimension. We denote the matrix of true labels and function evaluations stacked in the same manner as $\boldsymbol{\mathcal{Y}}_i$ and $\boldsymbol{f}(\boldsymbol{\mathcal{X}}_i)$, respectively. Here, $i$ denotes a client index $i \in \mathcal{C}$. Each tensor $\boldsymbol{\mathcal{J}}_i$, $\boldsymbol{\mathcal{Y}}_i$, and $\boldsymbol{f}(\boldsymbol{\mathcal{X}}_i)$ is a stacked representation of the data from each client and its neighbors. Explicitly, we have $\boldsymbol{\mathcal{J}}_i^{(k)} \in \mathbb{R}^{\tilde{N}_i \times d_2 \times d}$, $\boldsymbol{\mathcal{Y}}_i^{(k)} \in \mathbb{R}^{\tilde{N}_i \times d_2}$, and $\boldsymbol{f}(\boldsymbol{\mathcal{X}}_i) \in \mathbb{R}^{\tilde{N}_i \times d_2}$. $\tilde{N}_i$ denotes the total number of data points between client $i$ and its neighbors, and $d_2$ is the output dimension.

From here, each client performs the following operations to evolve its weights. **First**, compute the local NTK $\mathbf{H}_i^{(k)}$ from the Jacobian tensor $\boldsymbol{\mathcal{J}}_i^{(k)}$ using the definition of the NTK

$$\mathbf{H}_{i,mn}^{(k)} = \frac{1}{d_2} \langle \boldsymbol{\mathcal{J}}_i^{(k)}(\boldsymbol{x}_m), \boldsymbol{\mathcal{J}}_i^{(k)}(\boldsymbol{x}_n) \rangle_F. \quad (3)$$

Each element of the NTK is a pairwise Frobenius inner product between Jacobian matrices, where the indices $m$ and $n$ correspond to two different data points. **Second**, using $\mathbf{H}_i^{(k)}$, the client evolves their weights as follows (see Appendix C for more details):

$$\boldsymbol{f}^{(k,t)}(\boldsymbol{\mathcal{X}}_i) = (\mathbf{I} - e^{-\frac{\eta t}{N_i}\mathbf{H}_i^{(k)}})\boldsymbol{\mathcal{Y}}_i^{(k)} + e^{-\frac{\eta t}{N_i}\mathbf{H}_i^{(k)}}\boldsymbol{f}^{(k)}(\boldsymbol{\mathcal{X}}_i), \tag{4}$$

$$\boldsymbol{w}_i^{(k,t)} = \sum_{j=1}^{d_2}(\boldsymbol{\mathcal{J}}_{i,:j:}^{(k)})^\top \mathbf{R}_{i,:j}^{(k,t)} + \bar{\boldsymbol{w}}_i^{(k)}, \qquad \mathbf{R}_{i,:j}^{(k,t)} \equiv \frac{\eta}{\tilde{N}_i d_2}\sum_{u=0}^{t-1}[\boldsymbol{\mathcal{Y}}_i^{(k)} - \boldsymbol{f}^{(k,u)}(\boldsymbol{\mathcal{X}}_i)]. \tag{5}$$

**Third**, the client selects the weight $\boldsymbol{w}_i^{(k,t)}$ for a timestep $t$ with the lowest loss according to the evolved residual $\boldsymbol{f}^{(k,t)}(\boldsymbol{\mathcal{X}}_i) - \boldsymbol{\mathcal{Y}}_i$. This is used as the new weight $\boldsymbol{w}_i^{(k+1,0)}$ for the next communication round.

**Final Model Averaging**   Throughout the paper, we study the convergence of the aggregated model $\boldsymbol{w} = \frac{1}{M}\sum_{i=1}^{M} N_i \boldsymbol{w}_i$. In the decentralized setting, clients would average all models to create $\boldsymbol{w}$ after all training is completed. This may be done through a fully-connected topology, sequential averaging on a ring topology, or in a secure, centralized manner. Clients may also connect in a denser topology than that of training, and average with a desired number of neighbors. In practice, we observe that the aggregated model is more accurate than any individual client model. We study the impact of client averaging order on model performance with a client selection algorithm and show the results in Figure 5. Each client that opts in to model averaging contributes a portion of its data to a global validation set before training begins. Our client selection algorithm selects clients in the order of their accuracy on the validation set. We will demonstrate that in the practical setting, with a proper selection of clients, not all nodes must opt into final model averaging in order for the aggregate model to benefit from improved convergence. We note a difference between *model consensus*, often discussed in the DFL literature (Savazzi et al., 2020; Liu et al., 2022), and the proposed final model averaging approach. Model consensus refers to the gradual convergence of all client models to a single, unified model over numerous communication rounds. In contrast, our approach implements final model averaging as a distinct step performed after the completion of the training process.

Lastly, while memory efficiency is not the primary focus of this paper, we briefly note a technique to address potential memory constraints in NTK-DFL implementations. For scenarios involving dense networks or large datasets, we introduce Jacobian batching. This approach allows clients to process their local datasets in smaller batches, reducing memory complexity from $O(N_i d_2 d)$ to $O(N_i d_2 d/m)$, where $m$ is the number of batches. Clients compute and transmit Jacobians for each batch separately, evolving their weights multiple times per communication round. This complexity reduction allows clients to connect in a denser network for the same memory cost. A thorough discussion of network overhead can be found in Appendix D.

## 4 EXPERIMENTS

### 4.1 EXPERIMENTAL SETUP

**Datasets and Model Specifications**   Following Yue et al. (2022), we experiment on three datasets: Fashion-MNIST (Xiao et al., 2017), FEMNIST (Caldas et al., 2019), and MNIST (Lecun et al., 1998). Each dataset contains $C = 10$ output classes. For Fashion-MNIST and MNIST, data heterogeneity has been introduced in the form of non-IID partitions created by the symmetric Dirichlet distribution (Good, 1976). For each client, a vector $\boldsymbol{q}_i \sim \mathrm{Dir}(\alpha)$ is sampled, where $\boldsymbol{q}_i \in \mathbb{R}^C$ is confined to the $(C-1)$-standard simplex such that $\sum_{j=1}^{C} q_{ij} = 1$. This assigns a distribution over labels to each client, creating heterogeneity in the form of label-skewness. For smaller values of $\alpha$, clients possess a distribution concentrated over fewer classes. We test over a range of $\alpha$ values in order to simulate different degrees of heterogeneity. In FEMNIST, data is split

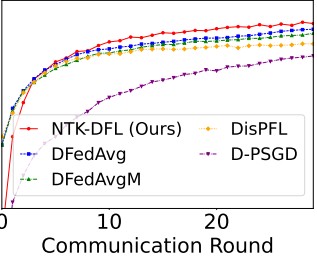

| Method | IID | $\alpha = 0.5$ | $\alpha = 0.1$ |
|--------|-----|----------------|----------------|
| DFedAvg | 18 | 31 | 83 |
| DFedAvgM | 23 | 43 | 86 |
| DisPFL | 43 | 87 | 200+ |
| D-PSGD | 53 | 79 | 125 |
| **NTK-DFL** | **12** | **17** | **18** |

Figure 2: Convergence of different methods on Fashion-MNIST for (left) highly non-IID with $\alpha = 0.1$ and (middle) IID settings. (Right) The table displays the communication rounds required to reach 85% test accuracy on Fashion-MNIST. We observe increased improvement in NTK-DFL convergence over baselines for more heterogeneous settings.

into shards based on the writer of each digit, introducing heterogeneity in the form of feature-skewness. For the model, we use a two-layer multilayer perceptron with a hidden width of 100 neurons for all trials.

**Network Topologies**    A sparse, time-variant $\kappa$-regular graph with $\kappa = 5$ was used as the standard topology for experimentation, where for each communication round $k$, a new random graph $\mathcal{G}^{(k)}$ with the same parameter $\kappa$ is created. Various values of $\kappa$ were tested to observe the effect of network density on model convergence. We also experimented with various topologies to ensure robustness to different connection settings. We used a network of 300 clients throughout our experiments.

**Baseline Methods**    We compare our approach to various state-of-the-art baselines in the DFL setting. These include D-PSGD (Lian et al., 2017), DFedAvg, DFedAvgM (Sun et al., 2023), and DisPFL (Dai et al., 2022). We also compare with the centralized baseline NTK-FL (Yue et al., 2022). The upper bound NTK-FL would consist of a client fraction of 1.0 where the server constructs an NTK from all client data each round, which is infeasible due to memory constraints. Instead, we conducted a comparison following Dai et al. (2022), with additional details regarding baselines and NTK-FL results found in Appendix B.

**Performance Metrics**    We evaluate the performance of the various DFL approaches by studying the aggregate model accuracy on a global, holdout test set. This ensures that we are measuring the generalization of the aggregate model from individual, heterogeneous local data to a more representative data sample. Our approach is in line with the goal of training a global model capable of improved generalization over any single local model (Section 3.1), unlike personalized federated learning where the goal is to fine-tune a global model to each local dataset (Tan et al., 2023). When evaluating the selection algorithm in Figure 5, we split the global test set in a 50:50 ratio of validation to test data. We use the validation data to sort the models based on their accuracy, and report the test accuracy in the figure.

## 4.2 EXPERIMENTAL RESULTS

**Test Accuracy & Convergence**    Our experiments demonstrate the superior convergence properties of NTK-DFL compared to baselines. Figure 2 illustrates the convergence trajectories of NTK-DFL and other baselines on Fashion-MNIST. We see that NTK-DFL convergence benefits are enhanced under increased heterogeneity. Under high heterogeneity with $\alpha = 0.1$, NTK-DFL establishes a 3–4% accuracy lead over the best-performing baseline within just five communication rounds and maintains this advantage throughout the training process. Additionally presented are the number of communication rounds necessary for convergence to 85% test accuracy, where NTK-DFL consistently outperforms all baselines. For the $\alpha = 0.1$ setting, NTK-DFL achieves convergence in 4.6 times fewer communication rounds than DFedAvg, the next best performing baseline. Figure 8 demonstrates a similar convergence advantage for NTK-DFL on both FEMNIST and non-IID MNIST datasets.

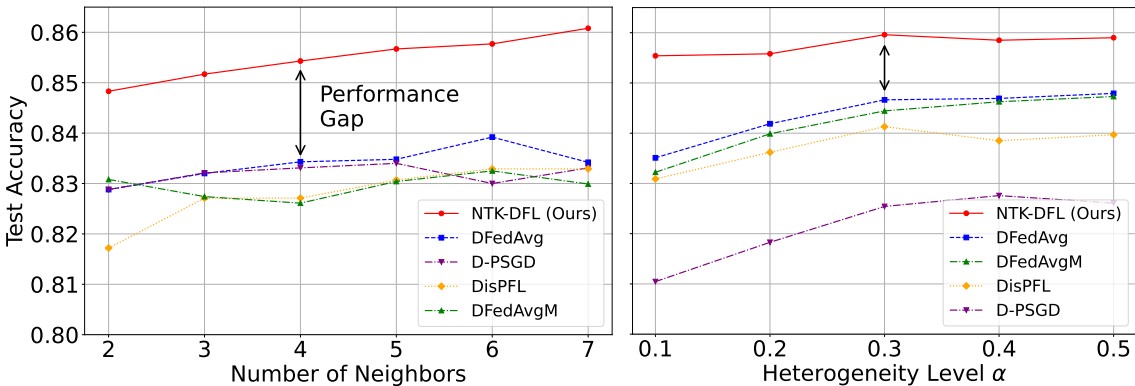

Figure 3: Performance of NTK-DFL vs. (left) sparsity level and (right) heterogeneity level (smaller $\alpha \rightarrow$ more heterogeneous). NTK-DFL outperforms the baselines and the gains are stable as the factors vary.

**Factor Analyses of NTK-DFL**  We evaluate NTK-DFL's performance over various factors, including the sparsity and heterogeneity levels, and the choices of the topology and weight initialization scheme. Figure 3 illustrates the test accuracy of NTK-DFL and other baselines as functions of the sparsity and heterogeneity levels, respectively. We observe a mild increase in convergence accuracy with decreasing sparsity. NTK-DFL experiences stable convergence across heterogeneity values $\alpha$ ranging from $0.1$ to $0.5$. The left plot reveals that NTK-DFL consistently outperforms baselines by 2–3% across all sparsity levels. The right plot demonstrates NTK-DFL's resilience to data heterogeneity—while baseline methods' performance deteriorates with decreasing $\alpha$, NTK-DFL maintains stable performance. Figure 9 illustrates the impact of network topology on NTK-DFL convergence. The dynamic topology accelerates convergence compared to the static topology, likely due to improved information flow among clients. Figure 10 demonstrates the effect of weight initialization on NTK-DFL performance. While random per-client initialization slightly slows convergence compared to uniform initialization, NTK-DFL exhibits robustness to these initialization differences.

**Gains Due to Final Model Aggregation**  Figure 4 demonstrates the dramatic effect of final model aggregation on final test accuracy. Though the individual client models decrease in accuracy as the level of heterogeneity increases, the final aggregated model remains consistent across all levels of heterogeneity (as seen in Figures 2 and 3). In the most heterogeneous setting $\alpha = 0.1$ that we tested, the difference between the mean accuracy of each client and the aggregated model accuracy is nearly 10% (see Figure 4). A similar phenomenon is observed as the client topology becomes more sparse. For the same heterogeneity setting with a sparser topology of $\kappa = 2$, the difference between these accuracies is nearly 15% (see Figure 12 in Appendix B). Though the individual performance of local client models may suffer under extreme conditions, the inter-client variance created by such unfavorable settings is exploited by model averaging to recuperate much of that lost performance.

Figure 7 suggests that inter-model variance enhances the performance of model averaging in DFL. While extreme

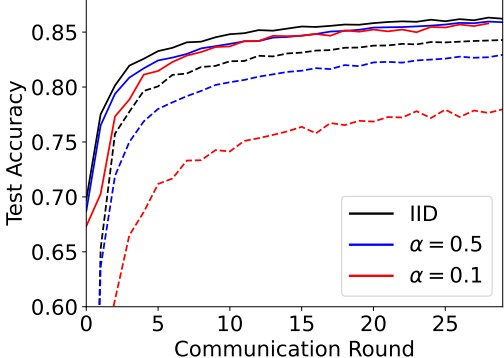

Figure 4: Performance gains of model averaging on convergence, trained on Fashion-MNIST. Solid lines correspond to the accuracy of the aggregated global model, whereas dotted lines correspond to the mean accuracy across client models. NTK-DFL's aggregated model maintains high performance, whereas mean client accuracy declines significantly with increased heterogeneity.

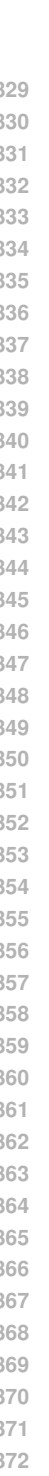
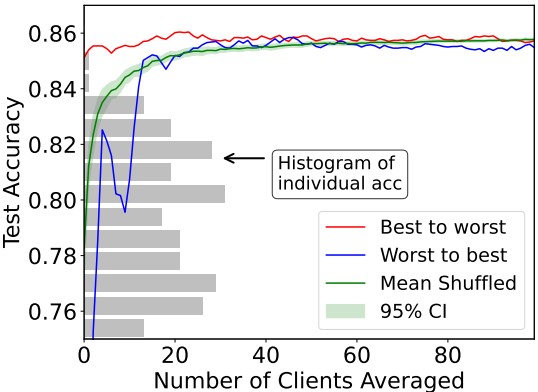
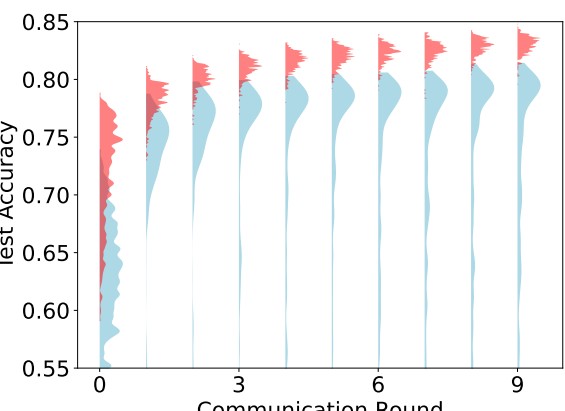

Figure 5: Final model test accuracy on Fashion-MNIST vs. the number of clients averaged for a highly heterogeneous setting with $\alpha = 0.1$. The histogram shows the distribution of individual client model accuracies. Three client selection criteria are tested: (red) high-to-low (proposed), (green) random, and (blue) low-to-high.

Figure 6: Distributions of individual client model accuracy vs. the communication round for Fashion-MNIST. The proposed scheme (in red) conducts per-round averaging among neighbors, whereas the ablated setup (in blue) does not. Per-round averaging significantly reduces the skewness of the distribution of model performance.

dissimilarity in model weights would likely result in poor performance of the averaged model, we observe that a moderate degree of variance can be beneficial. We posit that the NTK-based update steps generate a more advantageous level of variance compared to baseline approaches, contributing to improved overall performance.

**Selection Algorithm**    Figure 5 demonstrates the results of the selection algorithm for the final model aggregation. The selection algorithm is highly effective in the heterogeneous setting. The effect is most notable in the $\alpha = 0.1$ setting, where the performance with the selection algorithm significantly outperforms a random averaging order and the lower-bound averaging order. The proposed selection criterion requires the fewest clients to be averaged to achieve the same level of accuracy in this highly heterogeneous setting. In practical deployments, this has implications for final-round averaging in a fully decentralized setting. For example, clients may connect in a denser final topology and prioritize averaging with neighbors possessing a higher validation accuracy. This approach could optimize the efficacy of the final aggregation step while maintaining the decentralized nature of the system.

**Per-round Averaging Ablation Study**    In Figure 6, we perform an ablation study in which we remove the per-round parameter averaging that is a part of the NTK-DFL process. Here, clients forego the step of averaging their weight vectors with their neighbors during each commu-

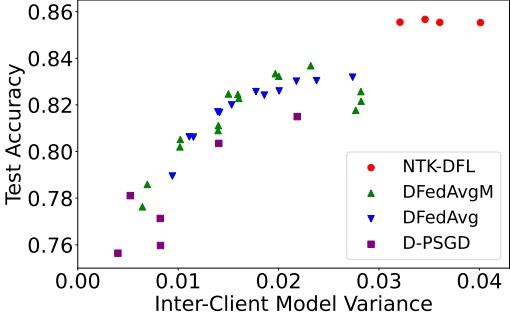

Figure 7: Relationship between model variance and final test accuracy on Fashion-MNIST. Each point represents a trial with distinct hyperparameters. The plot reveals a positive correlation between model variance and accuracy, suggesting that higher variance may benefit model averaging in DFL to a certain extent. Notably, the NTK-DFL approach demonstrates both higher accuracy and greater model variance compared to other methods.

nication round. Instead, clients compute Jacobians with respect to their original weight vector and send these to each of their neighbors (see Algorithm 3 in Appendix A). A massive distribution shift can be seen in the figure, where the distribution in the ablated setting is clearly skewed into lower accuracies. In contrast, NTK-DFL with per-round averaging demonstrates a much tighter distribution around a higher mean accuracy, effectively eliminating the long tail of low-performing models. Per-round averaging in NTK-DFL serves as a stabilizing mechanism against local model drift, safeguarding clients against convergence to suboptimal solutions early in the training process. In other words, client collaboration in the form of per-round averaging with neighbors ensures that no client lags behind in convergence. This is a particularly valuable feature in decentralized federated learning scenarios where maintaining uniformity across a diverse set of clients with heterogeneous data is a major challenge (Martínez Beltrán et al., 2023).

## 5 CONCLUSION AND FUTURE WORK

In this paper, we have introduced NTK-DFL, a novel approach to decentralized federated learning that leverages the neural tangent kernel to address the challenges of statistical heterogeneity in decentralized learning settings. Our work extends NTK-based training beyond centralized settings through novel studies in Jacobian batching and datapoint subsampling, while discovering a unique synergy between NTK evolution and decentralized model averaging that improves final model accuracy. Our method combines the expressiveness of NTK-based weight evolution with a decentralized architecture, allowing for efficient, collaborative learning without a central server. We reduce the number of communication rounds needed for convergence, which may prove advantageous for high-latency settings or those with heavy encoding/decoding costs.

There are promising unexplored directions for NTK-DFL. For instance, extending the algorithm to training models such as CNNs, ResNets (He et al., 2016), and transformers (Vaswani et al., 2017). Additionally, future research could explore the application of NTK-DFL to cross-silo federated learning scenarios, particularly in domains such as healthcare, where data privacy concerns and regulatory requirements often necessitate decentralized approaches. Lastly, NTK-DFL may serve as a useful paradigm for transfer learning applications in scenarios where a single, centralized source of both compute and data is not available.

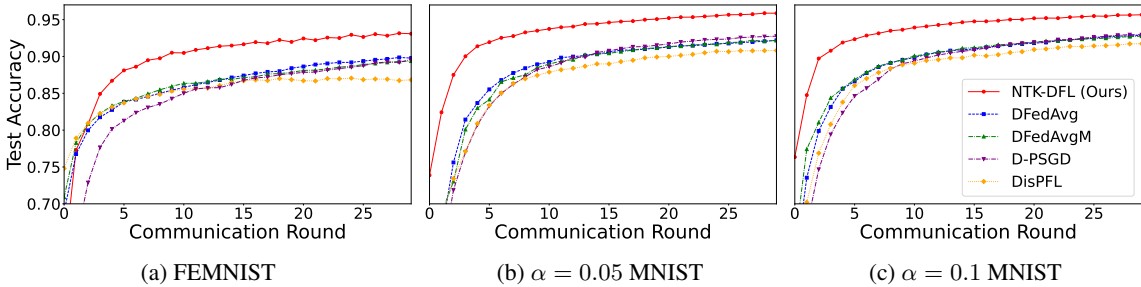

(a) FEMNIST          (b) $\alpha = 0.05$ MNIST          (c) $\alpha = 0.1$ MNIST

Figure 8: Convergence of various methods on (a) FEMNIST, (b) Non-IID MNIST ($\alpha = 0.05$), and (c) Non-IID MNIST ($\alpha = 0.1$). NTK-DFL consistently outperforms all baselines.

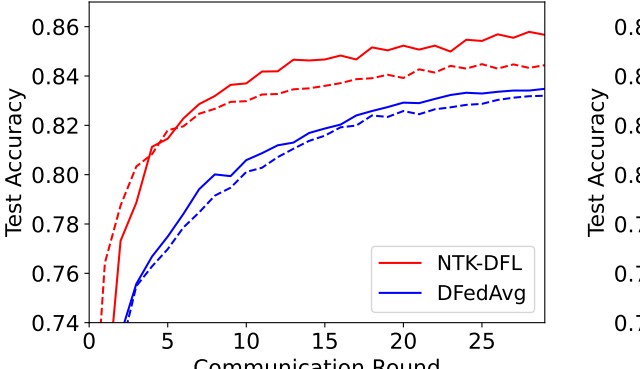
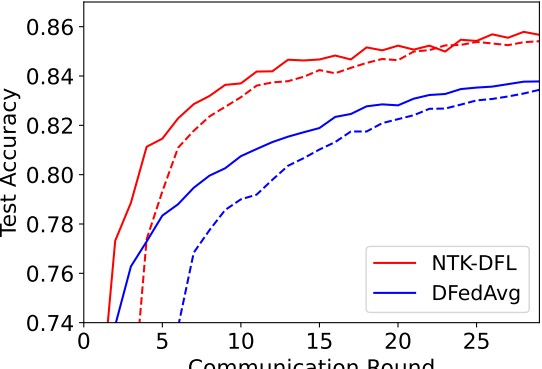

Figure 9: The effect of static vs. dynamic topology on NTK-DFL. Solid lines correspond to a dynamic topology, whereas dotted lines correspond to a static topology. Both methods benefit from the dynamic topology and NTK-DFL outperforms DFedAvg under both topologies. Other baselines are not drawn but perform similarly to DFedAvg.

Figure 10: The effect of different vs. identical weight initialization. Solid lines correspond to the same weight initialization for all clients, whereas dotted lines correspond to different initialization. The convergence of NTK-DFL is affected less than that of DFedAvg. Other baselines are not drawn but perform similarly to DFedAvg.

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

## A  NTK-DFL ALGORITHMS

---

**Algorithm 1** Consolidated Federated Learning Process

---

**Require:** A set of clients $\mathcal{C}$
1: Initialize weights $\boldsymbol{w}_i^{(0)}$ for each client $i$.
2: **for** each communication round $k = 1$ to $K$ **do**
3:   Initialize graph structure $G^{(k)} = (\mathcal{C}, E^{(k)})$, specifying the neighbors $\mathcal{N}_i^{(k)}$ for each client $i$.
4:   Execute Algorithm 2 for Per-Round Parameter Averaging
5:   Execute Algorithm 3 for Local Jacobian Computation and Sending
6:   Execute Algorithm 4 for Weight Evolution
7: **end for**

---

**Algorithm 2** Per-Round Parameter Averaging

---

**Require:** For each client $i$, a set of neighbors $\mathcal{N}_i^{(k)}$ and initial weights $\boldsymbol{w}_i^{(k)}$
1: **for** each client $i \in \mathcal{C}$ **in parallel do**
2:   Send $\boldsymbol{w}_i^{(k)}$ to all neighbors $j \in \mathcal{N}_i^{(k)}$
3:   Receive $\boldsymbol{w}_j^{(k)}$ from all neighbors $j \in \mathcal{N}_i^{(k)}$
4:   $\bar{\boldsymbol{w}}_i^{(k)} \leftarrow \frac{1}{|\mathcal{N}_i^{(k)}|+1}(\boldsymbol{w}_i^{(k)} + \sum_{j \in \mathcal{N}_i^{(k)}} \boldsymbol{w}_j^{(k)})$
5:   Send aggregated weight $\bar{\boldsymbol{w}}_i^{(k)}$ back to all neighbors $j \in \mathcal{N}_i^{(k)}$
6: **end for**

---

**Algorithm 3** Local Jacobian Computation and Sending Jacobians

---

**Require:** Each client $i$ knows its neighbors $\mathcal{N}_i^{(k)}$ and has access to local data $\mathbf{X}_i$ and the aggregated weights $\bar{\boldsymbol{w}}_j^{(k)}$ for each neighbor $j \in \mathcal{N}_i^{(k)}$.
1: **for** each client $i \in \mathcal{C}$ **in parallel do**
2:   Compute the Jacobian $\boldsymbol{J}_{i,i}^{(k)} \equiv \nabla_{\boldsymbol{w}} \boldsymbol{f}(\mathbf{X}_i; \bar{\boldsymbol{w}}_i^{(k)})$ using the client's own aggregated weight $\bar{\boldsymbol{w}}_i^{(k)}$ and local data $\mathbf{X}_i$.
3:   **for** each neighbor $j \in \mathcal{N}_i^{(k)}$ **do**
4:     Compute the Jacobian $\boldsymbol{J}_{i,j}^{(k)} \equiv \nabla_{\boldsymbol{w}} \boldsymbol{f}(\mathbf{X}_i; \bar{\boldsymbol{w}}_j^{(k)})$ using the neighbor's aggregated weight $\bar{\boldsymbol{w}}_j^{(k)}$ and client's local data $\mathbf{X}_i$.
5:     Send $\boldsymbol{J}_{i,j}^{(k)}$, true label $\mathbf{Y}_i$, and function evaluation $\boldsymbol{f}(\mathbf{X}_i; \bar{\boldsymbol{w}}_j^{(k)})$ to neighbor $j$.
6:   **end for**
7: **end for**

---

---

**Algorithm 4** Weight Evolution

---

**Require:** Each client $i$ has access to local data $\mathbf{X}_i$, initial weights $\bar{\boldsymbol{w}}_i^{(k)}$, and knows its neighbors $\mathcal{N}_i^{(k)}$

1: **for** each client $i \in \mathcal{C}$ after intra-client communication **do**
2:   Compute local Jacobian tensor $\boldsymbol{J}_{i,i}^{(k)}$ and receive $\boldsymbol{J}_{j,i}^{(k)}$ from each neighbor $j$
3:   Construct tensor $\boldsymbol{\mathcal{J}}_i^{(k)}$ from $\{\boldsymbol{J}_{ii}^{(k)}\} \cup \{\boldsymbol{J}_{ji}^{(k)} \mid j \in \mathcal{N}_i^{(k)}\}$
4:   Compute local NTK $\mathbf{H}_i^{(k)}$ using $\boldsymbol{\mathcal{J}}_i^{(k)}$:
5:   **for** each data point pair $(x_m, x_n)$ **do**
6:     $\mathbf{H}_{i,mn}^{(k)} \leftarrow \frac{1}{d_2}\langle \boldsymbol{\mathcal{J}}_i^{(k)}(x_m), \boldsymbol{\mathcal{J}}_i^{(k)}(x_n)\rangle_F$
7:   **end for**
8:   **for** each timestep $t = 1$ to $T$ **do**
9:     $\boldsymbol{f}^{(k,t)}(\boldsymbol{\mathcal{X}}_i) \leftarrow (\mathbf{I} - e^{\frac{\eta t}{N_i}\mathbf{H}_i^{(k)}})\boldsymbol{\mathcal{Y}}_i^{(k)} + e^{\frac{\eta t}{N_i}\mathbf{H}_i^{(k)}}\boldsymbol{f}^{(k)}(\boldsymbol{\mathcal{X}}_i)$
10:    $\boldsymbol{w}_i^{(k,t)} \leftarrow \sum_{j=1}^{d_2}(\boldsymbol{\mathcal{J}}_{i,:j:}^{(k)})^T \boldsymbol{R}_{i,:j}^{(k,t)} + \bar{\boldsymbol{w}}_i^{(k)}$
11:   **end for**
12:   Select $\boldsymbol{w}_i^{(k+1,0)} \leftarrow \boldsymbol{w}_i^{(k,t)}$ with the lowest loss given the residual $(\boldsymbol{f}^{(k,t)}(\boldsymbol{\mathcal{X}}_i) - \boldsymbol{\mathcal{Y}}_i)$
13: **end for**

---

# B  ADDITIONAL EXPERIMENTAL DETAILS

## B.1  BASELINES

NTK-FL is the only centralized baseline that we compare with. We choose a part rate that ensures that the busiest node in the centralized setting is no busier than the busiest decentralized setting. By busier, we mean the degree of the node or number of clients communicating with it. We note that NTK-FL is not an upper bound in this case due to the comparison being founded on node busyness. Evaluating NTK-FL in the same setting as the table in Figure 2, NTK-FL converges to threshold accuracy in 73, 85, and 180 communication rounds for heterogeneity settings IID, $\alpha = 0.5$, and $\alpha = 0.1$ respectively. D-PSGD is one of the first decentralized, parallel algorithms for distributed machine learning that allows nodes to only communicate with neighbors. DFedAvg adapts FedAvg to the decentralized setting, and DFedAvgM makes the use of SGD-based momentum and extends DFedAvg. Both use multiple local epochs between communication rounds, like vanilla FedAvg. DisPFL is a personalized federated learning approach that aims to train a global model and personalize it to each client with a local mask. In order to make the comparision fair, we report the accuracy of the global model on our test set.

## B.2  HYPERPARAMETERS

We perform a hyperparameter search over each baseline and select the hyperparameters corresponding to the best test accuracy. We use the $\alpha = 0.1$ Fashion-MNIST test accuracy at communication round 30 as the metric for selection. This is done because the majority of comparisons take place on Fashion-MNIST in the non-IID setting. For D-PSGD, we use a learning rate of $0.1$, and a batch size of 10 (local epochs are defined to be one in this approach). For DFedAvg, we use a learning rate of $0.1$, a batch size of 25, and 20 local epochs. For DFedAvgM, we use a learning rate of $0.01$, a batch size of 50, 20 local epochs, and a momentum of $0.9$. For DisPFL, we use a learning rate of $0.1$, a batch size of 10, and 10 local epochs. Following (Dai et al., 2022), we use the sparsity rate of $0.5$ for DisPFL. As for the NTK-DFL, we use a learning rate of $0.01$ and search over values $t \in \{100, 200, \ldots, 800\}$ during the weight evolution process.

### B.3 INTER-MODEL VARIANCE

We describe the inter-model variance between NTK-DFL clients in Figure 7. Here, the variance is calculated as follows

$$V = \frac{1}{d_2} \sum_{j=1}^{d_2} \sqrt{\sum_{i=1}^{M} [\bar{\boldsymbol{w}} - \boldsymbol{w}_i]_j^2} \tag{6}$$

where we investigate the average per-parameter variance to normalize for the scale of each parameter among clients.

### B.4 FURTHER EXPERIMENTAL RESULTS

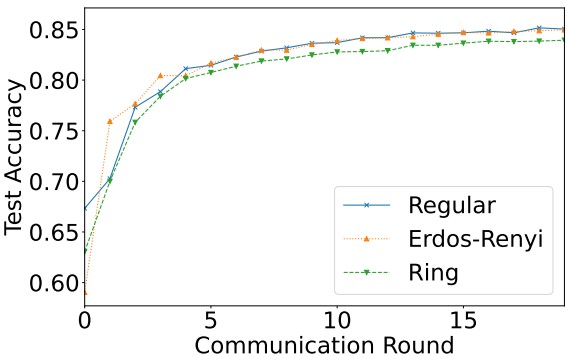

Figure 11: Convergence of NTK-DFL across different dynamic topologies, trained on Fashion-MNIST. NTK-DFL is evaluated with (blue) a $\kappa = 5$ regular graph, (yellow) an Erdos-Renyi random graph with five mean neighbors, and (green) a ring topology, where each client is connected to two neighbors. We observe that NTK-DFL demonstrates steady convergence across different topology classes.

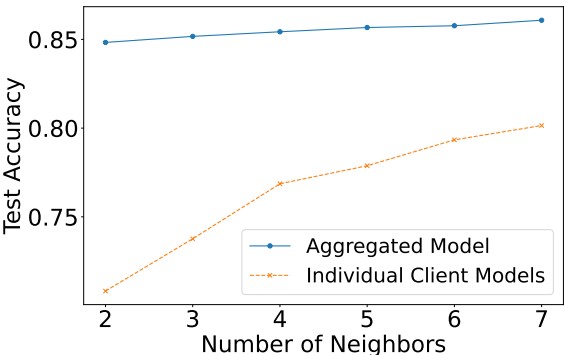

Figure 12: NTK-DFL model accuracy as a function of neighbor count ($\kappa$), trained on Fashion-MNIST. Notably, (blue) the aggregated model accuracy across NTK-DFL clients remains consistent, even as network sparsity varies. This stability persists despite a significant decline in (yellow) mean individual client test accuracy as the number of neighbors decreases.

PLEASE NOTE: All sections (Appendix D and E) and figures below this line are new. We leave unhighlighted for sake of readability.

## C  ADDITIONAL DETAILS ON WEIGHT EVOLUTION

In implementation, computing the matrix exponential $e^{-\frac{\eta t}{N_i}\mathbf{H}_i^{(k)}}$ in Equation 4 to evolve weights can be computationally expensive. In practice, the weights are evolved according to the more general differential equation from which Equation 4 is derived, reliant upon the linearized model approximation $\boldsymbol{f}(\boldsymbol{\mathcal{X}_i};\bar{\boldsymbol{w}}_j^{(k,t)}) \approx \boldsymbol{f}(\boldsymbol{\mathcal{X}_i};\bar{\boldsymbol{w}}_j^{(k,0)}) + \nabla_{\boldsymbol{w}}\boldsymbol{f}(\boldsymbol{\mathcal{X}_i};\bar{\boldsymbol{w}}_j^{(k,0)})(\bar{\boldsymbol{w}}_j^{(k,t)} - \bar{\boldsymbol{w}}_j^{(k,0)})$. The differential equation is as follows

$$\frac{d}{dt}\boldsymbol{f}(\boldsymbol{\mathcal{X}_i};\bar{\boldsymbol{w}}_j^{(k,t)}) = -\eta\mathbf{H}_j^{(k)}\nabla_{\boldsymbol{f}}\mathcal{L}$$

Here, $\mathcal{L}$ is the loss function. For example, for a half mean-squared error (MSE) loss, term on the right becomes the residual matrix $\nabla_{\boldsymbol{f}}\mathcal{L} = \boldsymbol{f}(\boldsymbol{\mathcal{X}_i};\bar{\boldsymbol{w}}_j^{(k,t)}) - \boldsymbol{\mathcal{Y}}_i$. During weight evolution, a client $j$ evolves their neighboring function evaluation from the initial condition $\boldsymbol{f}(\boldsymbol{\mathcal{X}_i};\bar{\boldsymbol{w}}_j^{(k,0)})$ to the time-evolved $\boldsymbol{f}(\boldsymbol{\mathcal{X}_i};\bar{\boldsymbol{w}}_j^{(k,t)})$ using a differential equation solver and the differential equation above. To implement Equation 5, we use a process similar to Yue et al. (2022) where the initial client residual is evolved over a series of timesteps specified by the user. For user-specified timesteps, the loss at that time is found using the evolved residual. Then, the best-performing weights are evolved using the left side of Equation 5 and selected for the next communication round.

## D  DISCUSSION OF NETWORK OVERHEAD

While analysis of memory and communication overhead are not a central theme of this paper, we include strategies to mitigate both forms of overhead for practical deployment. A thorough analysis of optimization and parallelization is out of the scope of this work and we leave it to future research.

### D.1  JACOBIAN BATCHING

We introduce Jacobian batching to address potential memory constraints in NTK-DFL implementations. For scenarios involving dense networks or large datasets, clients can process their local datasets in smaller batches, reducing memory complexity from $O(N_i d_2 d)$ to $O(N_i d_2 d/m_1)$, where $m_1$ is the number of batches. Clients compute and transmit Jacobians for each batch separately, evolving their weights multiple times per communication round. This approach effectively trades a single large NTK $\mathbf{H} \in \mathbb{R}^{N \times N}$ for $m_1$ smaller NTKs $\mathbf{H}_{m_1} \in \mathbb{R}^{N/m_1 \times N/m_1}$ that form block diagonals of $\mathbf{H}$, where $N$ represents the total number of data points between client $i$ and its neighbors $\mathcal{N}_i$. While some information is lost in the uncomputed off-diagonal entries of $\mathbf{H}$, this is mitigated by the increased frequency of NTK evolution steps. Figure 13 demonstrates this phenomenon, where an increasing batch number $m_1$ actually leads to improved convergence. This complexity reduction enables clients to connect in a denser network for the same memory cost.

### D.2  COMMUNICATION COST

Compared to traditional weight-based approaches that communicate a client's parameters $\mathbf{w}_i$ each round, NTK-DFL utilizes Jacobian matrices to enhance convergence speed and heterogeneity resilience. This tensor has memory complexity $O(N_i d_2 d)$, where $N_i$ denotes the number of data points between client $i$ and its neighbors $\mathcal{N}_i$, $d$ is the model parameter dimension, and $d_2$ is the output dimension. We propose the following

strategies to improve the communication efficiency of NTK-DFL while maintaining convergence properties in heterogeneous settings.

**Data Subsampling**   We introduce an approach where clients sample a $1/m_2$ fraction of their data each round for NTK evolution. Clients follow the protocol described in Section 3.2, but exchange Jacobian matrices of reduced size. As demonstrated in Figure 14, moderate values of $m$ yield light performance degradation, validating this communication reduction strategy.

**Jacobian Compression**   We employ several techniques to reduce Jacobian tensor dimensionality. First, we apply top-$k$ sparsification, zeroing out elements with the smallest magnitude (Alistarh et al., 2018). The remaining non-zero values are quantized to $b$ bits. Additionally, we introduce a shared random projection matrix $\mathbf{P} \in \mathbb{R}^{d_1 \times d_1'}$ generated from a common seed, creating projections $\mathbf{Z}_i = \mathbf{X}_i \mathbf{P}$ that reduce input dimension from $d_1$ to $d_1'$. This combination of techniques maintains convergence properties while significantly reducing communication costs. Note that similar compression schemes applied to weight-based approaches lead to significant degradation in performance (Yue et al., 2022). Figure 15 illustrates the relative differences in communication load for a different combinations of the techniques above, with a sparsification of 0.5, quantization to 6 bits, a sampling of $m_2 = 4$, and a projection to $d_1' = 200$ for the full optimization curve. In Figure 16, we see the communication comparison of NTK-DFL updates with less expressive, DFedAvg weight updates. The communication-optimized NTK-DFL converges in 3.9 times fewer communication rounds compared to DFedAvg (19 rounds for NTK-DFL vs. 75 rounds for DFedAvg). However, with more expressive updates than DFedAvg, it uses 7.5 times as many bits (195 MB for NTK-DFL vs. 26 MB for DFedAvg). This enforces the idea that NTK-DFL is especially useful in scenarios where convergence in few communication rounds is important, such as those with non-negligible encoding and decoding delays.

# E   RECONSTRUCTION ATTACK

While privacy preservation is not the primary focus of this work, we conduct a brief analysis of data privacy in NTK-DFL. Following the reconstruction attack method of Zhu et al. (2019), we evaluate the feasibility of reconstructing client data from transmitted Jacobian matrices under varying compression levels. Our experiments range from basic top-$k$ sparsification with sparsity $0.25$ to combined sparsification with random projection to dimension $d_1' = 200$. Figure 17 illustrates that client data reconstruction becomes increasingly difficult when a random projection is additionally applied to the Jacobian matrices.

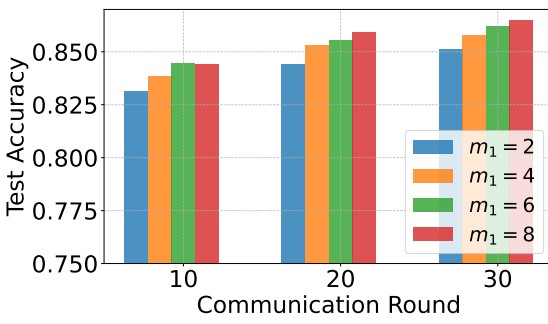

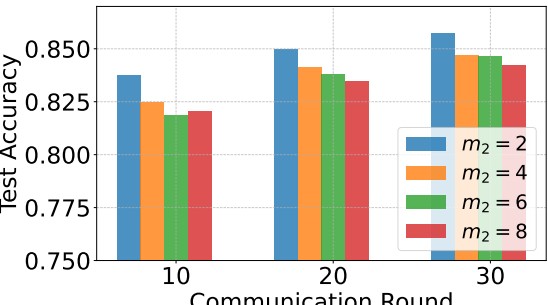

Figure 13: Test accuracy of NTK-DFL vs. communication round for various Jacobian batch numbers $m_1$, with higher $m_1$ values denoting more batches (Fashion-MNIST, $\alpha = 0.1$). We observe a general, counterintuitive increase in test accuracy with an increased number of batches.

Figure 14: Test accuracy of NTK-DFL vs. communication round for sampling divisors $m_2$. Different from Jacobian batching, only a $1/m_2$ fraction of client data is selected each communication round. We observe a slight decrease in test accuracy with increased $m_2$.

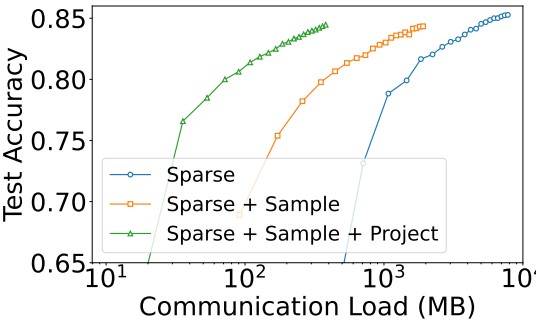

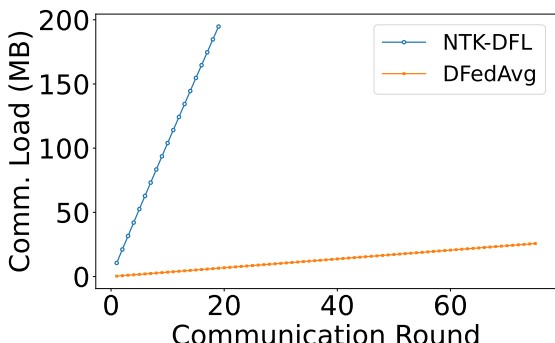

Figure 15: Comparison of NTK-DFL variants with progressive communication optimizations. Data sampling and projection technique provides compounding reductions in communication load compared to sparsification alone, while the fully optimized variant demonstrates significantly lower communication requirements at a comparable test accuracy.

Figure 16: Comparison of communication trade-off between NTK-DFL and DFedAvg across accuracy thresholds. While NTK-DFL achieves convergence in fewer communication rounds than DFedAvg, its more expressive parameter updates require a higher communication volume per round.

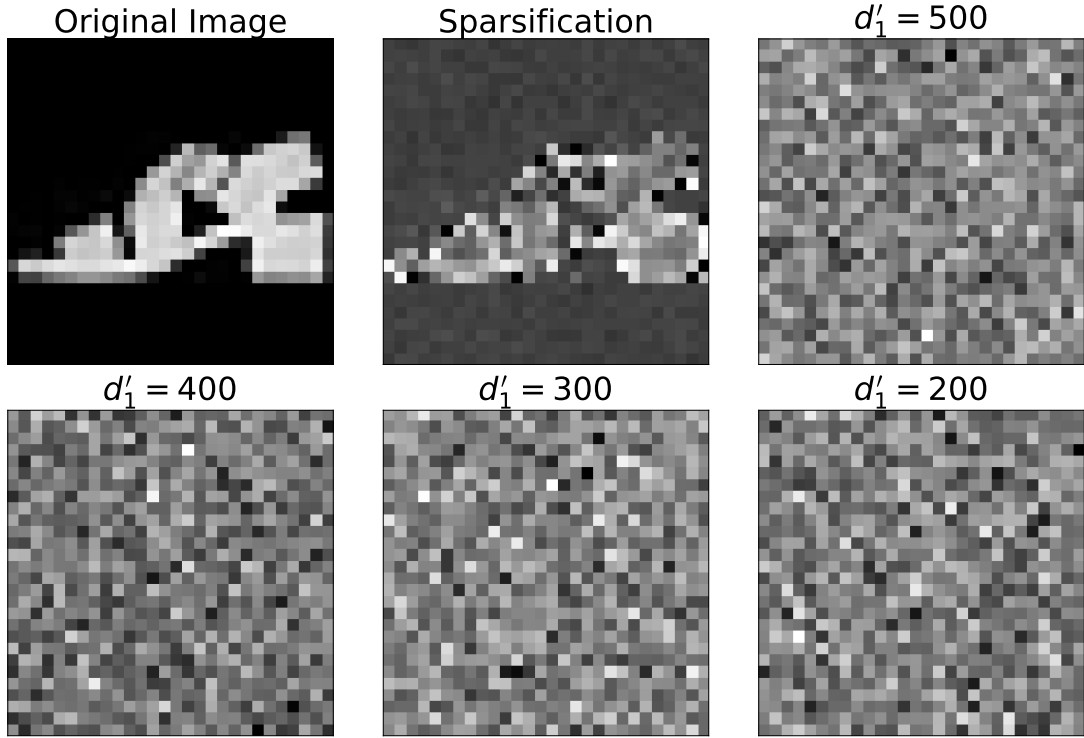

Figure 17: Reconstruction attack of client data from Jacobian matrices for various levels of compression. For the image corresponding to sparsified matrices (the middle image of the first row), no random projection is done. We observe the ability to reconstruct a very noisy version of client data. For the other images, we use sparsification and a random projection to dimension $d'_1$. We observe an inability to reconstruct client data when the random projection is additionally applied.

