# OpenReview forum: "NTK-DFL: Enhancing Decentralized Federated Learning in Heterogeneous Settings via Neural Tangent Kernel"
_ICLR.cc/2025/Conference — Submitted to ICLR 2025_

### Official Review · Reviewer_Ump1 · 2024-11-01

**Soundness:** 2
**Presentation:** 2
**Contribution:** 3
**Rating:** 5
**Confidence:** 3

**Summary:**

The paper proposes to leverage the Neural Tangent Kernel (NTK) to guide decentralized federated learning. More specifically, in each communication round, clients send their weights to neighbors and ask neighbors to calculate the Jacobian on neighbors' datasets. Then, clients receive the Jacobian from neighbors, and construct the empirical NTK matrix with the data from all neighbors. Next, each client updates the weights with reconstructed NTK. The proposed method NTK-DFL is shown to outperform standard DFL algorithms on heterogeneous datasets.

**Strengths:**

The paper is motivated by an important problem in decentralized federated learning: heterogeneity. The idea to reconstruct empirical NTK from neighbors is novel and interesting. Experiments also show great potential of the NTK-DFL in tacking heterogeneous datasets.

**Weaknesses:**

- My major concern is the communication overhead of NTK-DFL. Conventional decentralized gradient descent only communicates gradients with neighbors, so the communication cost is $O(d)$. However, NTK-DFL needs to communicate Jacobian with the cost of $O(N_i\times d)$, where $N_i$ is the number of data in one client. This approach does not seem scalable to big datasets. Authors mentioned Jacobian batching in line 216. It would be interesting to see the trade-off between communication cost and performance for different batch sizes.

- The presentation of weight evolution can be made clearer. It seems negative signs are missing in the exponents in (4).

**Questions:**

- How are implement (4) and (5) implemented in practice? To exactly calculate the exponential map, eigen-decomposition on $\tilde{N}_i\times\tilde{N}_i$ matrix $H$ is needed. It would be better to explicitly present the implementation in practice.

- Can authors also present the communication cost in terms of bits rather than communication rounds?

- Can the method scale to larger scale problems like TinyImagenet?

---

> ### Author Response · Authors · 2024-11-24
> **Response to Reviewer 4 (Ump1) Part 1**
>
> **Comment 1:** My major concern is the communication overhead of NTK-DFL. Conventional decentralized gradient descent only communicates gradients with neighbors, so the communication cost is $O(d)$. However, NTK-DFL needs to communicate Jacobian with the cost of $O(N_i \times d)$, where $N_i$ is the number of data in one client. This approach does not seem scalable to big datasets. Authors mentioned Jacobian batching in line 216. It would be interesting to see the trade-off between communication cost and performance for different batch sizes.
>
> **Response:** We thank the reviewers for raising the question about communication overhead in NTK-DFL. We have added a section in Appendix D.1 on Jacobian batching, and refer the reviewer to our newly added section on datapoint sampling and compression (Appendix D.2). Jacobian batching would induce the same communication cost but reduce memory complexity, so we instead describe our strategies to mitigate communication overhead.
>
> In the updated manuscript, we present a study on the effect of datapoint sampling on convergence in Figure 14. The datapoint sampling would reduce the communication each round by a factor of $m_2$, as Jacobian matrices scale directly with the number of datapoints. We also introduce various methods to reduce communication load, including previously mentioned subsampling, quantization, top-$k$ sparsification, and random projection of input data. A more thorough description is provided in Appendix D (we refer the reviewers to the revised manuscript due to the excessive amount of added content) and the effect on communication load is presented in Figure 16.
>
> In addition, we have added a description in Appendix D to make explicit the consideration of communication cost and introduce our various methods of mitigation:
>
> > “Compared to traditional weight-based approaches that communicate a client’s parameters $\mathbf{w}_i$ each round, NTK-DFL utilizes Jacobian matrices to enhance convergence speed and heterogeneity resilience. This tensor has memory complexity $O(\tilde{N_i}d_2d)$, where $\tilde{N_i}$ denotes the number of data points between client $i$ and its neighbors, $d$ is the model parameter dimension, and $d_2$ is the output dimension. We propose the following strategies to improve the communication efficiency of NTK-DFL while maintaining convergence properties in heterogeneous settings…”
>
> **Comment 2:** The presentation of weight evolution can be made clearer. It seems negative signs are missing in the exponents in (4). How are implement (4) and (5) implemented in practice? To exactly calculate the exponential map, eigen-decomposition on $N_i\times N_i$ matrix $H$ is needed. It would be better to explicitly present the implementation in practice.
>
> **Response:** We thank the reviewers for pointing out this mistake. We have added the negative signs to the exponents in Eq. (4). We have added Appendix C, which makes explicit the weight evolution process and how it is implemented. In practice, the weights are evolved using a differential equation solver (Chen, 2018) according to the more general differential equation (shown below) from which Eq. (4) is derived:
> $$
> \frac{d}{dt} f\left(\mathbf{X}_i; \bar{\mathbf{w}}_j^{(k, t)}\right)
> = -\eta \mathbf{H}_j^{(k)} \nabla_f \mathcal{L}
> $$
>
> To implement Eq. (5), the client evolves the initial residual matrix over a series of timesteps specified by the user. For user-specified timesteps, the loss at that time is found using the evolved residual. Then, the best-performing weights are calculated using the left side of Eq. (5) (shown below) and selected for the next communication round.
> $$w_i^{(k,t)} = \sum_{j=1}^{d_2} (J_{i,j}^k)^T R_{i,j}^{k,t} + \bar{w}_i^k$$
> (We apologize for not bolding the $w_i$ above, markdown isn't cooperating with that equation).
>
> **Comment 3:** Can authors also present the communication cost in terms of bits rather than communication rounds?
>
> **Response:**
> We have added the new results in terms of bits in Figure 16 (Appendix E). The NTK-DFL approach is inherently more communication-intensive per communication rounds, as it sends more expressive Jacobian matrices rather than model weights or gradient updates. Through the use of compression techniques including subsampling client data, sparsification, and random projection (newly added Appendix D.2), we are able to reduce the communication load of NTK-DFL by a factor of ~300.

---

> ### Author Response · Authors · 2024-11-24
> **Response to Reviewer 4 (Ump1) Part 2**
>
> **Comment 4:** Can the method scale to larger scale problems like TinyImagenet?
>
> **Response:** We thank the reviewers for their concern about scalability. In the revised manuscript, we have preliminarily explored a pathway toward scaling to larger datasets like Tiny ImageNet via Jacobian batching and datapoint subsampling, but a thorough analysis will be deferred to future studies. In general, the scaling of Jacobian matrices with model size and the number of output classes makes complex datasets more difficult due to memory overhead. While our current results demonstrate NTK-DFL's advantages for heterogeneous and decentralized training, we have added an explicit discussion of memory constraints along with initial approaches to address them. The newly added discussion of Jacobian batching is included below:
>
> > "We introduce Jacobian batching to address potential memory constraints in NTK-DFL implementations. For scenarios involving dense networks or large datasets, clients can process their local datasets in smaller batches, reducing memory complexity from $O(\tilde{N_i}d_2d)$ to $O(\tilde{N_i}d_2d/m_1)$, where $m_1$ is the number of batches. Clients compute and transmit Jacobians for each batch separately, evolving their weights multiple times per communication round. This approach effectively trades a single large NTK $H \in \mathbb{R}^{\tilde{N_i}\times\tilde{N_i}}$ for $m_1$ smaller NTKs $H_{m_1}\in\mathbb{R}^{\tilde{N_i}/{m_1}\times \tilde{N_i}/{m_1}}$ that form block diagonals of $H$, where $\tilde{N}_i$ represents the total number of data points between client i and its neighbors. While some information is lost in the uncomputed off-diagonal entries of $H$, this is mitigated by the increased frequency of NTK evolution steps. This complexity reduction enables clients to connect in a denser network for the same memory cost. Figure 13 demonstrates this phenomenon, where an increasing batch number $m_1$ actually leads to improved convergence. This complexity reduction enables clients to connect in a denser network for the same memory cost.”
>
> **References:**
>
> Chen, R. (2018). Torchdiffeq: PyTorch Implementation of Differentiable ODE Solvers. https://github.com/rtqichen/torchdiffeq. Accessed November 24, 2024

---

### Official Review · Reviewer_wkMx · 2024-11-02

**Soundness:** 3
**Presentation:** 2
**Contribution:** 2
**Rating:** 5
**Confidence:** 3

**Summary:**

This work proposes an approach leveraging the NTK to train client models in the decentralized setting while introducing a synergy between NTK-based evolution and model averaging. It exploits intermodel variance and improves both accuracy and convergence in heterogeneous settings. The empirical results show the proposed approach consistently achieves higher accuracy than baselines in highly heterogeneous settings and reaches target performance in fewer communication rounds with multiple datasets, network topologies, and heterogeneity settings.

**Strengths:**

1.The experiments demonstrate that the proposed method is useful.
2.The study with NTK approach on decentralized FL is meaningful.

**Weaknesses:**

1. A significant concern revolves around the novelty of the proposed method. It seems that the proposed method may appear to be an extension of NTK from FL to DFL with some effective trick methods. It is essential for the authors to underscore their distinctive contributions in a more prominent manner.
2. In terms of experimental baselines, it is recommended that the authors include the most recent decentralized federated learning method (DFedSAM (Shi et al. (2023)) for a comprehensive comparison. This will enhance the paper's completeness and relevance in the context of the current state of the field.
3.The analysis in line 310-325 “Gains Due to Final Model Aggregation” is not clear enough. Where is the 10% in line 318 and 15% in line 321?

**Questions:**

1.See the weakness above.

---

> ### Author Response · Authors · 2024-11-24
> **Response to Reviewer 3 (wkMx) Part 1**
>
> **Comment 1**: A significant concern revolves around the novelty of the proposed method. It seems that the proposed method may appear to be an extension of NTK from FL to DFL with some effective trick methods. It is essential for the authors to underscore their distinctive contributions in a more prominent manner.
>
> **Response**: We appreciate the opportunity to clarify the novel aspects of NTK-DFL. While our work builds on NTK-FL, we introduce several key innovations in the context of DFL. We added a short section in the conclusion to emphasize these contributions:
>
> > "Our work extends NTK-based training beyond centralized settings through novel studies in Jacobian batching and datapoint subsampling, while discovering a unique synergy between NTK evolution and decentralized model averaging that improves final model accuracy."
>
> We also edited the third contribution bullet to highlight our results on the Jacobian matrix compression:
>
> > "This aggregated model exhibits robust performance across various network topologies, datasets, data distributions, and compression measures."
>
> Lastly, we add the following sentences to the Introduction to highlight Jacobian compression results and reconstruction-attack results which were added in the new manuscript:
>
> > "We demonstrate that NTK-DFL maintains high performance even under aggressive compression measures. Through reconstruction attack studies, we also analyze how this compression affects data privacy."
>
> To summarize, in the updated manuscript:
> * We added Jacobian batching results in Appendix D.1 to address the memory constraints unique to decentralized settings, allowing flexible memory-computation trade-offs when clients have limited resources.
> * We added the study of the effects of datapoint subsampling on convergence in Appendix D.2, which allows for the management of communication load and memory complexity.
>
> To emphasize unique contributions from the original manuscript:
> * We discover and exploit a unique synergy between NTK evolution and decentralized model averaging - the heterogeneity in client models creates beneficial variance that can improve final model accuracy.
> * We provide extensive analysis on how topology, data sampling, and model initialization affect NTK evolution in decentralized settings - insights not available from centralized approaches and not present in the work by Yue et al. (2022).
>
> **Comment 2**: In terms of experimental baselines, it is recommended that the authors include the most recent decentralized federated learning method (DFedSAM (Shi et al. (2023)) for a comprehensive comparison. This will enhance the paper's completeness and relevance in the context of the current state of the field.
>
> **Response**: Thank you for your suggestion to include DFedSAM (Shi et al., 2023) as an additional baseline. While their code is not currently available for a fair and comprehensive comparison, we agree that comparing against DFedSAM would provide valuable insights once the authors release their code.
>
> **Comment 3**: The analysis in line 310-325 "Gains Due to Final Model Aggregation" is not clear enough. Where is the 10% in line 318 and 15% in line 321?
>
> **Response**: We thank the reviews for their careful attention to this section. In the revised manuscript, we have added explicit references to the figures which display the accuracy differences mentioned in the text.
>
> > "The difference between the mean accuracy of each client and the aggregated model accuracy is nearly 10% (see Figure 4)."
>
> > "The difference between these accuracies is nearly 15% (see Figure 12 in Appendix B)."
>
> The 10% in line 318 comes from the difference between dotted and solid red curves in Figure 4, which shows the aggregated client accuracy versus the communication round. The 15% in line 321 comes from the difference between the blue and yellow points for $\kappa=2$ in Figure 12 (Appendix B).
>
> **References**
>
> Yifan Shi, Li Shen, Kang Wei, Yan Sun, Bo Yuan, Xueqian Wang, and Dacheng Tao. Improving the model consistency of decentralized federated learning. In Proceedings of the 40th International Conference on Machine Learning, 2023.
>
> Kai Yue, Richeng Jin, Ryan Pilgrim, Chau-Wai Wong, Dror Baron, and Huaiyu Dai. Neural tangent kernel empowered federated learning. In Proceedings of the 39th International Conference on Machine Learning, volume 162 of Proceedings of Machine Learning Research, pp. 25783–25803. PMLR, 17–23 Jul 2022.

---

> > ### Comment · Reviewer_wkMx · 2024-11-25
> > **Comments**
> >
> > Thank you for your response.  After careful consideration, I decided to keep my score.
> > Some other suggestions:
> > 1. If you would like to add the comparisons with DFedSAM, the code of SAM optimizer can be found at https://github.com/YMJS-Irfan/DP-FedSAM/tree/master/fedml_api/dpfedsam from the same author.
> > 2. The revision looks better than the last version. It would be fine if the layout of the last two pages could be improved.

---

> > > ### Author Response · Authors · 2024-11-25
> > >
> > > We appreciate the reviewer's suggestions. We noticed that the provided codebase appears to implement a centralized federated learning approach. Could you please help us identify which specific implementation you're referencing regarding the DFedSam algorithm in *Improving the Model Consistency of Decentralized Federated Learning*?

---

> > > > ### Comment · Reviewer_wkMx · 2024-11-26
> > > > **Suggestions**
> > > >
> > > > 1. **The core difference for DFedAvg and DFedSAM** is the optimizer with SGD and SAM: Line 53 and 55 at https://github.com/YMJS-Irfan/DP-FedSAM/blob/master/fedml_api/dpfedsam/my_model_trainer.py
> > > >
> > > > 2. Other local training differences could be seen in the comparisons of dp-fedsam and dpsgd at https://github.com/YMJS-Irfan/DP-FedSAM/blob/master/fedml_api/dpfedsam/my_model_trainer.py   and    https://github.com/rong-dai/DisPFL/blob/master/fedml_api/standalone/dpsgd/my_model_trainer.py  from dispfl code base.   Maybe you should understand some details about SAM in the paper of DFedSAM or *Generalized Federated Learning via Sharpness Aware Minimization* to avoid mistakes.
> > > >
> > > > 3. Combine SAM functions at https://github.com/YMJS-Irfan/DP-FedSAM/blob/master/fedml_api/dpfedsam/sam.py, you can finish the whole local training progress.  Then, for other things, you could follow the code as you do for DFedAvg.

---

> > > > > ### Author Response · Authors · 2024-11-30
> > > > >
> > > > > We thank the reviewers for providing these detailed implementation resources. While a comparison against DFedSAM would be valuable, it would require substantial adaptation beyond the provided materials - particularly since these implementations are for centralized, differentially private federated learning rather than the decentralized setting. A fair comparison would require carefully implementing the approach's intricacies and conducting a thorough hyperparameter search. Given these scope constraints, we will leave this comparison to future work.

---

### Official Review · Reviewer_45hy · 2024-11-03

**Soundness:** 3
**Presentation:** 3
**Contribution:** 3
**Rating:** 6
**Confidence:** 4

**Summary:**

This paper introduce Neural Tangent Kernal into Decentralized Federated Learning, method of which is called NTK-DFL. The corresponding framework, NTK-DFL weight evolving method, empirically works on 3 common vision datasets in federated learning literatures.

**Strengths:**

- Presentation is easy to read.
- Straightforward motivation and clear validation: NTK works well in Centralized FL; it's not used in Decentralized FL yet; a vanilla method NTK-DFL is proposed; it empirically works well.
- Experimental results about comparison and ablation seem good.

**Weaknesses:**

- Privacy concerned. In the 3rd step of proposed framework NTK-DFL, ground truth label are exchanged among clients, which would lead to more serious bad impacts. This potention privacy leakage breaks the claim made in Line 45-48. More discussion about privacy preserving should be considered.
- Additional overhead. centralized federated learning system do have a heavy bandwidth issue on the path from clients to the server. However, the additional ones in NTK-DFL exchanged on the entire network system is far larger than that of centralized federated learning system. Further analyses on network overhead should be involved and make the paper more solid.
- The content and the main claim are supported by the experiment. However, more experiments will be considered for higher rating. More detailed ablation study, analyses on overhead through the whole DFL system and NTK on other network topologies and protocols (e.g., multi-server) are recommended.

**Questions:**

- How to prevent privacy data leakage in NTK-DFL's 3rd step? Is there any technique to enhance this? For example more discussion on differential privacy.
- What's the comparison to other methods about network overhead? Is there a comparison of total bytes transferred per round for NTK-DFL v.s. centralized approaches, or an analysis of how the overhead scales with number of clients and model size.
-  How about other network topologies and protocols in DFL?

---

> ### Author Response · Authors · 2024-11-24
> **Response to Reviewer 2 (45hy) Part 1**
>
> **Comment 1:** Privacy concerned. In the 3rd step of proposed framework NTK-DFL, ground truth label are exchanged among clients, which would lead to more serious bad impacts. This potention privacy leakage breaks the claim made in Line 45-48. More discussion about privacy preserving should be considered.
>
> **Response:** We thank the reviewer for this privacy concern. In the revised manuscript, we perform a  reconstruction-attack study in which clients have access to the ground truth label, which is different from a typical reconstruction attack where the label is inferred through gradient matching (Zhu et al., 2019). While having access to ground truth labels does increase potential privacy risks, our study shows that compression/dimension reduction defenses including Jacobian sparsification and random projection can help mitigate these risks.
> We have added reconstruction attack results in Figure 17 of Appendix E, demonstrating that compressed Jacobians provide some protection against data reconstruction, even with the inclusion of the ground truth label. (Note that our new results in Figure 15 of Appendix D.2 show that aggressive compression maintains relative performance while reducing information shared between clients.) While a comprehensive privacy analysis is beyond our current scope, NTK-DFL represents a point on the spectrum between utility and privacy - trading some privacy for expressiveness compared to weight-sharing approaches.
>
> **Comment 2:** Additional overhead. centralized federated learning system do have a heavy bandwidth issue on the path from clients to the server. However, the additional ones in NTK-DFL exchanged on the entire network system is far larger than that of centralized federated learning system. Further analyses on network overhead should be involved and make the paper more solid.
>
> **Response:** We thank the reviewer for this important concern. We have added a more detailed analysis of NTK-DFL's overhead in Appendix D.2, including results showing how subsampling, quantization, and random projection can reduce communication costs by a factor of ~300x. Additionally, our Jacobian batching protocol (newly added results shown in Figure 13) offers users a flexible trade-off between memory complexity and time complexity. We note that in most DFL approaches, total communication volume scales with the number of neighbors per client, which can lead to greater communication volume than centralized FL. A comprehensive analysis of network overhead involves many factors, which we plan to leave for future studies.
>
> **Comment 3:** The content and the main claim are supported by the experiment. However, more experiments will be considered for higher rating. More detailed ablation study, analyses on overhead through the whole DFL system and NTK on other network topologies and protocols (e.g., multi-server) are recommended.
>
> **Response:** We thank the reviewer for these suggestions. In the revised manuscript, we have added comprehensive studies on Jacobian batching (Appendix D.1) and data subsampling (Appendix D.2) to address memory and communication complexity, respectively. We also add experimental results on communication complexity after integrating the compression of Jacobian matrices through sparsification and random projection of input data (Appendix D.2).
> The original submission includes topology analysis - Figure 9 compares static vs dynamic topologies and Figure 10 examines weight initialization effects. Furthermore, results in Figure 11 (Appendix B) compare NTK-DFL across regular, Erdos-Renyi (mean degree 5), and ring topologies. While performance naturally decreases with fewer neighbors (e.g., ring topology with 2 neighbors), NTK-DFL maintains its core benefits across different network structures. While multi-server protocols are outside our current focus and expertise, we followed works like DFedAvg (Sun et al., 2021) and DFedSam (Shi et al., 2023) in analyzing convergence properties and key factors affecting decentralized training performance.

---

> ### Author Response · Authors · 2024-11-24
> **Response to Reviewer 2 (45hy) Part 2**
>
> **Comment 5:**
> What's the comparison to other methods about network overhead? Is there a comparison of total bytes transferred per round for NTK-DFL v.s. centralized approaches, or an analysis of how the overhead scales with number of clients and model size.
>
> **Response:**  We value the reviewer's concern about network overhead. In the revised manuscript, we have added a comparison with the next-most performant DFL approach, DFedAvg. Our result emphasizing the trade-off between communication volume and communication rounds is included below (Figure 16, Appendix D). Decentralized federated learning approaches often generate more total communication volume than their centralized counterparts, as all clients communicate with their neighbors each round rather than being selected to participate using a part rate. Because of this, we intentionally compare with other decentralized approaches.
>
> In NTK-DFL, both communication and memory overhead scale with the number of data points (and thus implicitly with the number of clients) and model size. This communication scaling is fundamentally similar to weight-sharing approaches like DFedAvg, where overhead scales with both client count and model size. We include an explicit discussion of communication overhead in Appendix D.2:
>
> > "Compared to traditional weight-based approaches that communicate a client's parameters $\mathbf{w}_i$ each round, NTK-DFL utilizes Jacobian matrices to enhance convergence speed and heterogeneity resilience. This tensor has memory complexity $O(N_id_2d)$, where $N_i$ denotes the number of data points between client $i$ and its neighbors $N_i$, $d$ is the model parameter dimension, and $d_2$ is the output dimension. We propose the following strategies to improve the communication efficiency of NTK-DFL while maintaining convergence properties in heterogeneous settings…"
>
> To address the comparison with centralized approaches, we used a comparison following Dai et al. (2022) to compare with the centralized NTK-FL (Yue et al., 2022). These results were described in Appendix B in the original manuscript:
>
> > "We choose a part rate that ensures that the busiest node in the centralized setting is no busier than the busiest decentralized setting. By busy, we mean the degree of the node or number of clients communicating with it. We note that NTK-FL is not an upper bound in this case due to the comparison being founded on node busyness. Evaluating NTK-FL in the same setting as the table in Figure 2, NTK-FL converges to threshold accuracy in 73, 85, and 180 communication rounds for heterogeneity settings IID, α = 0.5, and α = 0.1 respectively."
>
> **References**
>
> Rong Dai, Li Shen, Fengxiang He, Xinmei Tian, and Dacheng Tao. DisPFL: Towards communication-efficient personalized federated learning via decentralized sparse training, 2022.
>
> Tao Sun, Dongsheng Li, and Bao Wang. Decentralized federated averaging. IEEE Transactions on Pattern Analysis and Machine Intelligence, 45(4):4289–4301, 2021.
>
> Kai Yue, Richeng Jin, Ryan Pilgrim, Chau-Wai Wong, Dror Baron, and Huaiyu Dai. Neural tangent kernel empowered federated learning. In Proceedings of the 39th International Conference on Machine Learning, volume 162 of Proceedings of Machine Learning Research, pp. 25783–25803. PMLR, 17–23 Jul 2022.
>
> Ligeng Zhu, Zhijian Liu, and Song Han. Deep leakage from gradients. In Advances in Neural Information Processing Systems, volume 32, 2019.

---

> ### Comment · Reviewer_45hy · 2024-11-26
>
> Most of the issues I raised were resolved, mostly on privacy issues but the author doesn't intend to answer my questions head on right in this work. Instead, showing that it doesn't attack successfully by gradient matching.
> My question now is: does this algorithm transfer the ground truth label in the 3rd Line?
> After reading other reviewers' queries as well as additional material, code, and continuing to see if the authors address the rest of the questions, I decided to keep my rating (for now).

---

> > ### Author Response · Authors · 2024-11-27
> >
> > We thank the reviewer for careful attention to the additional material and code. To answer the reviewer's question explicitly: Yes, the ground truth label $\mathbf{Y}_i$ is shared between clients. We reference Section 3.2 of the manuscript:
> >
> > > "Each client sends every neighbor their respective Jacobian tensor $\mathbf{J}_{i,j}^{(k)}$, true label $\mathbf{Y}_i$,
> > and function evaluation $\mathbf{f(X_i; \bar{w}^{(k)}_j )}$."

---

### Official Review · Reviewer_ob3c · 2024-11-04

**Soundness:** 2
**Presentation:** 2
**Contribution:** 3
**Rating:** 5
**Confidence:** 4

**Summary:**

This paper proposes a novel framework for decentralized federated learning (DFL) that leverages a Neural Tangent Kernel (NTK)-based update mechanism instead of typical gradient descent methods. Specifically, clients send labels and Jacobians to their neighbors, who then use tools from NTK to obtain the trained neural network instead of relying on gradient descent. The paper provides empirical results validating that this approach significantly outperforms previous baselines in highly heterogeneous settings, achieving 4.6 times fewer communication rounds.

**Strengths:**

- The idea of using the NTK paradigm for DFL without gradient descent is both interesting and novel.
- This paper contributes a practical algorithm for DFL.

**Weaknesses:**

- The proposed method requires clients to share their respective Jacobians, true labels, and function evaluations with their neighbors. This seems to violate the privacy-preserving feature of FL. More discussion is needed.
- The experiments were only validated on simple datasets (MNIST, Fashion-MNIST, and EMNIST); it is necessary to test on more complex datasets, such as CIFAR-100.
- The notations could be improved to make the method and algorithm clearer.

**Questions:**

1. The derivative notation should be: $\boldsymbol{J}_{i, j}^{(k)} \equiv\big[\nabla _{\bar{w}_j} \boldsymbol{f} (\mathbf{X}_i; \overline{\boldsymbol{w}} _j^{(k)}) \big]^{\top} \Rightarrow$

$
\boldsymbol{J}_{i, j}^{(k)} \equiv\big[\nabla _{w} \boldsymbol{f}(\mathbf{X}_i ; \overline{\boldsymbol{w}}_j^{(k)})\big]^{\top}
$

2. The expression “A global or aggregated model may take the form $\boldsymbol{w}=\frac{1}{M} \sum _{i=1}^M N_i \boldsymbol{w}_i$” should be revised to: $\boldsymbol{w}=\frac{1}{N} \sum _{i=1}^M N_i \boldsymbol{w}_i$, where $N=\sum _{i=1}^M N_i.$

3. Consider simplifying the set of clients using the notation $\mathcal{C}=${$1,..i,..M$} instead of $\mathcal{C}=${$C_1,...C_M$}. This way, subsequent neighborhoods can be expressed as $\mathcal{N}_i^{(k)}= ${$ j \mid(i, j) \in E^{(k)}$ }, making later expressions more concise.



4. The true label $Y_i$ in Figure 1 does not match the definition $\mathbf{Y}_i$ in the main text.

5. Given the definition of the aggregated model as $\boldsymbol{w}=\frac{1}{N} \sum_{i=1}^M N_i \boldsymbol{w}_i$, Eq. (1) should be modified to:
   $
   \overline{\boldsymbol{w}}_i^{(k)}=\frac{1}{N_i+\sum _{j \in \mathcal{N}_i^{(k)}}N_j}(N_i\boldsymbol{w}_i^{(k)}+\sum _{j \in \mathcal{N}_i^{(k)}} N_j\boldsymbol{w}_j^{(k)}).
   $

6. It is currently unclear how NTK-DFL differs from the previous work by Yue et al. (2022); it seems to extend their work to decentralized FL. It would be helpful for the paper to add more discussion on the unique aspects of NTK-DFL in the context of DFL.

7. NTK-DFL requires each client to share local true labels, function evaluations, Jacobians, and other information with their neighbors. This appears to contradict the fundamental privacy-preserving features of FL. More discussion on privacy protection is recommended.

**Details Of Ethics Concerns:**

N.A.

---

> ### Author Response · Authors · 2024-11-24
> **Response to Reviewer 1 (ob3c) Part 1**
>
> **Comment 1**: NTK-DFL requires each client to share local true labels, function evaluations, Jacobians, and other information with their neighbors. This appears to contradict the fundamental privacy-preserving features of FL. More discussion on privacy protection is recommended.
>
> **Response**: We thank the reviewer for raising this important concern. Our revised manuscript has added a preliminary reconstruction-attack study with the help of compression/dimension reduction defenses including Jacobian sparsification and random projection. We've added reconstruction attack results in Figure 17 of Appendix E, demonstrating that compressed Jacobians provide some protection against data reconstruction. (Note that our new results in Figure 15 of Appendix D.2 show that aggressive compression maintains relative performance while reducing information shared between clients.) While a comprehensive privacy analysis is beyond our current scope, NTK-DFL represents a point on the spectrum between utility and privacy - trading some privacy for expressiveness compared to weight-sharing approaches.
>
> **Comment 2**: The experiments were only validated on simple datasets (MNIST, Fashion-MNIST, and EMNIST); it is necessary to test on more complex datasets, such as CIFAR-100.
>
> **Response**: We appreciate this important point and we acknowledge this limitation. In the revised manuscript, we have preliminarily explored a pathway toward scaling to larger datasets via Jacobian batching and datapoint subsampling, but a thorough analysis will be deferred to future studies. In general, the scaling of Jacobian matrices with model size and the number of output classes makes complex datasets more difficult due to memory overhead. While our current results demonstrate NTK-DFL's advantages for heterogeneous and decentralized training, we have added an explicit discussion of memory constraints along with initial approaches to address them.
>
> **Comment 3**:
> * The derivative notation should be:
>
> $
> J_{i,j}^{(k)} \equiv [\nabla_{\bar{w}_j}f(X_i; \bar{w}_j^{(k)})]^\top  \Rightarrow
> $
>
> $
> J_{i,j}^{(k)} \equiv \left[ \nabla_w f(\mathbf{X}_i; \bar{w}_j^{(k)}) \right]^\top
> $
>
> * The expression "A global or aggregated model may take the form..." should be revised to:
>   $w = \frac{1}{M}\sum_{i=1}^M N_iw_i \Rightarrow w = \frac{1}{N}\sum_{i=1}^M N_iw_i$
>
> * Consider simplifying the set of clients using the notation $\mathcal{C} = \\{i, ..., 1, ..., M\\}$ instead of $\\{C_1, ..., C_m\\}$. This way, subsequent neighborhoods can be expressed as $\mathcal{N}_i^{(k)}=\\{j | (i,j)\in E^{(k)}\\}$, making later expressions more concise.
>
> * The true label $Y_i$ in Figure 1 does not match the definition in the main text.
>
> * Given the definition of the aggregated model as $\\bar{\\mathbf{w}} = \frac{1}{N}\sum_{i=1}^M N_i\\bar{\\mathbf{w}}_i$, Eq. (1) should be modified to:
> \\(\\bar{\\mathbf{w}}_i^{(k)} = \\frac{1}{N_i+\\sum_j N_j} (N_i \\bar{\\mathbf{w}}_i^{(k)} + \\sum_j N_j\\bar{\\mathbf{w}}_j^{(k)})\\) (the sum is over $j\\in\\mathcal{N}_i^{(k)}$, but markdown formatting wasn't cooperating above)
>
> **Response**: We thank the reviewer for carefully following the mathematics and providing corrections and suggested improvements. We have corrected the derivative notation, global model definition, client set notation, and Eq. (1) as suggested. We have also modified Figure 1 to ensure consistency with the notation in the main text.
>
> **Comment 4**: It is currently unclear how NTK-DFL differs from the previous work by Yue et al. (2022); it seems to extend their work to decentralized FL. It would be helpful for the paper to add more discussion on the unique aspects of NTK-DFL in the context of DFL.
>
> **Response**: We appreciate the opportunity to clarify the novel aspects of NTK-DFL. While our work builds on NTK-FL, we introduce several key innovations in the context of DFL. We added a short section in the conclusion to emphasize these contributions:
>
> > "Our work extends NTK-based training beyond centralized settings through novel studies in Jacobian batching and datapoint subsampling, while discovering a unique synergy between NTK evolution and decentralized model averaging that improves final model accuracy."
>
> We also edited the third contribution bullet to highlight our results on the Jacobian matrix compression:
>
> > "This aggregated model exhibits robust performance across various network topologies, datasets, data distributions, and compression measures."
>
> Lastly, we add the following sentences to the Introduction to highlight Jacobian compression results and reconstruction-attack results which were added in the new manuscript:
>
> > "We demonstrate that NTK-DFL maintains high performance even under aggressive compression measures. Through reconstruction attack studies, we also analyze how this compression affects data privacy."

---

> > ### Author Response · Authors · 2024-11-24
> > **Responses to Reviewer 1 (ob3c) Part 2**
> >
> > To summarize, in the updated manuscript:
> > * We added Jacobian batching results in Appendix D.1 to address the memory constraints unique to decentralized settings, allowing flexible memory-computation trade-offs when clients have limited resources.
> > * We added the study of the effects of datapoint subsampling on convergence in Appendix D.2, which allows for the management of communication load and memory complexity.
> >
> > To emphasize unique contributions from the original manuscript:
> > * We discover and exploit a unique synergy between NTK evolution and decentralized model averaging - the heterogeneity in client models creates beneficial variance that can improve final model accuracy.
> > * We provide extensive analysis on how topology, data sampling, and model initialization affect NTK evolution in decentralized settings - insights not available from centralized approaches and not present in the work by Yue et al. (2022).
> >
> > **References:**\
> > Kai Yue, Richeng Jin, Ryan Pilgrim, Chau-Wai Wong, Dror Baron, and Huaiyu Dai. Neural tangent kernel empowered federated learning. In Proceedings of the 39th International Conference on Machine Learning, volume 162 of Proceedings of Machine Learning Research, pp. 25783–25803. PMLR, 17–23 Jul 2022.

---

> > > ### Comment · Reviewer_ob3c · 2024-11-29
> > >
> > > Thank you for your detailed response. After considering the feedback from other reviewers, I have decided to raise my score.

---

### Author Response · Authors · 2024-11-15
**NTK-DFL Code**

Hello! Here is the link for the NTK-DFL code. The README has a description of prerequisites for the code and example usage.

https://anonymous.4open.science/r/ntk-dfl-07DF/

---

### Meta-Review · Area_Chair_bbS4 · 2024-12-19

**Metareview:**

The authors introduce a novel framework for decentralized federated learning (DFL) which replaces the typical gradient descent method with a Neural Tangent Kernel (NTK)-based update mechanism. Particularly, clients send labels and Jacobians to their neighbors, who will leverage NTK to obtain the trained neural net. This is an interesting contribution and the idea of using NTK-based update instead of gradient descent for DFL is interesting and novel.

However, the reviewers raised critical concerns regarding (1) potential violation of privacy concern when clients have to share labels and Jacobians; (2) lack of experiments on more complicated datasets and comparison with important baselines.

The authors have made substantial efforts to address the concerns raised by the reviewers but have not provided any extra experiments following multiple reviewers' suggestions. Some of the suggested baseline comparisons, as pointed out by Reviewer wkMx and Ump1, are important, as acknowledged by the authors.

The reason stated by the authors (in response to wkMx) for not providing comparison during rebuttal is that it would require intricate implementation. While I understand that the authors might not have enough time to do that, it cannot be used as an excuse because without those comparison, the empirical studies are not sufficiently conclusive. This basically means this paper needs more time and polishing to reach an acceptable state.

Thus, unfortunately, lacking strong empirical studies, I cannot recommend acceptance for this paper. The authors are encouraged to strengthen the paper's empirical studies following the suggestions from the reviewers for future submission.

**Additional Comments On Reviewer Discussion:**

The reviewers have responded to the authors' rebuttal, which have addressed some concerns. But, the rebuttal was not able to provide any additional empirical comparisons as requested by the reviewers.

Consequently, the authors have not been able to convincingly address some of key concerns regarding comparison with important baselines. The ratings therefore remain in the borderline with not enough enthusiasm for acceptance.

---

### Decision · Program_Chairs · 2025-01-22

Reject